# Similarities and Differences between Multisystem Inflammatory Syndrome in Children (MIS-C) and Kawasaki Disease Shock Syndrome

**DOI:** 10.3390/children10091527

**Published:** 2023-09-08

**Authors:** Jin Lee, Beom Joon Kim, Kyoung-Soon Cho, Jung Woo Rhim, Soo-Young Lee, Dae Chul Jeong

**Affiliations:** 1Department of Pediatrics, College of Medicine, The Catholic University of Korea, Seoul 06591, Republic of Korea; pedleejin@naver.com (J.L.); beomjoon.kim@catholic.ac.kr (B.J.K.); soon926@hanmail.net (K.-S.C.); jwrhim@catholic.ac.kr (J.W.R.); dcjeong@catholic.ac.kr (D.C.J.); 2Department of Pediatrics, Incheon St. Mary’s Hospital, The Catholic University of Korea, Incheon 21431, Republic of Korea; 3Department of Pediatrics, Eunpyeong St. Mary’s Hospital, The Catholic University of Korea, Seoul 03312, Republic of Korea; 4Department of Pediatrics, Bucheon St. Mary’s Hospital, The Catholic University of Korea, Bucheon 14647, Republic of Korea; 5Department of Pediatrics, Daejeon St. Mary’s Hospital, The Catholic University of Korea, Daejeon 34943, Republic of Korea; 6The Vaccine Bio Research Institute, College of Medicine, The Catholic University of Korea, Seoul 06591, Republic of Korea; 7Department of Pediatrics, Seoul St. Mary’s Hospital, The Catholic University of Korea, Seoul 06591, Republic of Korea

**Keywords:** multisystem inflammatory syndrome in children, Kawasaki disease shock syndrome, Kawasaki disease

## Abstract

This study aimed to investigate the characteristics of COVID-19-associated multisystem inflammatory syndrome in children (MIS-C) and Kawasaki disease shock syndrome (KDSS) and to compare the similarities and differences between the two diseases. The incidence of KDSS and MIS-C was also estimated. Medical records of patients diagnosed with MIS-C or KDSS at four hospitals from January 2013 to December 2022 were retrospectively reviewed. Thirty-one patients were enrolled in the study in either an MIS-C group (*n* = 22) or a KDSS group (*n* = 9). The incidence of KDSS in KD was 0.8% (9/1095) and the incidence of MIS-C versus KD was 10.2% (22/216). Compared with the MIS-C group, the KDSS group had longer hospital stays and more severe systemic inflammation (e.g., anemia, elevated C-reactive protein, hypoalbuminemia, and pyuria) and organ dysfunction (e.g., number of involved organs, shock, vasoactive infusion, and intensive care unit admission). All patients in the MIS-C group, but none in the KDSS group, including two patients during the COVID-19 pandemic, had laboratory evidence of SARS-CoV-2 infection. MIS-C and KDSS shared demographic, clinical, and laboratory characteristics; organ dysfunction; treatment; and outcomes. Overall severity was more severe in patients with KDSS than in those with MIS-C. The most important difference between MIS-C and KDSS was whether SARS-CoV-2 was identified as an infectious trigger.

## 1. Introduction

Most children with COVID-19, which is caused by infection with severe acute respiratory syndrome coronavirus 2 (SARS-CoV-2), are asymptomatic or have mild symptoms [1,2]. However, in April 2020, Verdoni et al. [3] reported an outbreak in Italian children presenting with a severe systemic inflammatory disease resembling Kawasaki disease (KD) who often required admission to an intensive care unit (ICU) due to cardiac dysfunction and shock. Soon after, similar cases began to be reported in the United Kingdom, Spain, France, the United States, India, Brazil, and many other countries, including Korea [2,4]. The condition is now called COVID-19-associated multisystem inflammatory syndrome in children (MIS-C) or pediatric inflammatory multisystem syndrome temporarily related to SARS-CoV-2 (PIMS-TS) [4,5,6]. MIS-C is a serious complication that can occur two to eight weeks after SARS-CoV-2 infection and is characterized by infectious triggers, systemic inflammation, and organ dysfunction [7]. The presence of “KD-like features” in patients with MIS-C has received clinical attention, but the extent of organ dysfunction in patients with MIS-C is of greater importance as it directly affects the disease course and outcome [8,9,10].

Organ dysfunction, including shock, is also seen in a unique and severe form of KD called Kawasaki disease shock syndrome (KDSS) [11,12]. The incidence of KDSS ranges from 1% to 7% in patients with KD [13]. Like MIS-C, KDSS has a relatively short history [14]. The term was coined in 2009 by Kanegaye et al. [15] to differentiate this severe form from hemodynamically stable KD. In fact, KDSS is distinct in various aspects [16,17,18]. Coronary artery abnormalities (CAAs) and intravenous immunoglobulin (IVIG) resistance, a major concern in the field of KD research, are more common in KDSS than in hemodynamically stable KD [12]. KDSS also has severe manifestations in terms of systemic inflammation (e.g., cytopenia, elevated C-reactive protein [CRP], or hypoalbuminemia) and organ dysfunction (e.g., cardiac, gastrointestinal, or neurologic manifestations); all features are encountered in MIS-C [17]. Because MIS-C and KD share KD-like features, many studies have been conducted to elucidate the relationship between MIS-C and KD [8,9]. Few studies have addressed the relationship between MIS-C and KDSS [10,14], even though MIS-C and KDSS share organ dysfunction in addition to KD-like features. 

In this study, we investigated the characteristics of patients diagnosed with MIS-C or KDSS and compared the similarities and differences. We also estimated the incidence of KDSS and MIS-C in Korea, which had not previously been reported.

## 2. Materials and Methods

### 2.1. Study Design and Population

This was a multicenter retrospective study of patients who were hospitalized with MIS-C or KDSS between January 2013 and December 2022, at the Departments of Pediatrics of four hospitals in Korea: Seoul St. Mary’s Hospital, Eunpyeong St. Mary’s Hospital, Bucheon St. Mary’s Hospital, and Daejeon St. Mary’s Hospital. All data were collected through the Clinical Data Warehouse (CDW), which incorporates affiliated hospitals under the Catholic Medical Center in Korea. The CDW collects and distributes clinical data to researchers, including more than 15 million anonymized electronic medical records, following a comprehensive ethical board review [19].

The present study population was selected from the CDW. Patients who had a diagnosis code for MIS-C (U10.9) and met the MIS-C case definition [7] were assigned to the MIS-C group. Among patients with KD (M30.3), those who received vasoactive drugs or had a secondary diagnosis code for hypotension (I95.9) were allocated to the KDSS group. The study protocol was approved by the Institutional Review Board of The Catholic University of Korea (approval number: HC23WIDI0040), and the need for informed consent was waived due to the study’s retrospective nature.

### 2.2. Definitions

MIS-C was diagnosed using the Council of State and Territorial Epidemiologists/Centers for Disease Control and Disease Prevention (CSTE/CDC) case definition [7]: (1) febrile patient aged <21 years; (2) evidence of systemic inflammation; (3) new onset of organ dysfunction in ≥2 of 5 categories (cardiac, shock, mucocutaneous, gastrointestinal, or hematologic); and (4) laboratory evidence of SARS-CoV-2 infection. 

A diagnosis of KD is based on the American Heart Association criteria [20,21]. Complete KD was diagnosed by the presence of fever and ≥4 of 5 KD-like features: rash, conjunctivitis, oropharyngeal inflammation, extremity changes, and cervical lymphadenopathy. Incomplete KD was indicated by the presence of fever and <4 of 5 KD-like features plus compatible laboratory or echocardiographic features. Patients who also had shock or hypotension (systolic hypotension for age, a decrease in systolic blood pressure from baseline ≥ 20%, or clinical signs of hypoperfusion) were diagnosed with KDSS [15,16].

As assessment of IVIG resistance or initial treatment failure was made when patients with MIS-C or KDSS received second-line treatment due to persistent fever ≥ 36 h after completion of initial treatment (IVIG ± systemic steroids) [22]. A diagnosis of CAA was based on the z-score classification system recommended in the American Heart Association guidelines: a z-score for coronary artery size ≥ 2.0 indicated a coronary artery dilatation and a z-score for coronary artery size ≥ 2.5 indicated a coronary artery aneurysm [20].

### 2.3. Data Collections

Clinical data were obtained for age, sex, height, weight, duration of fever at admission, length of hospital stay, date of MIS-C or KDSS diagnosis, five KD-like features, complete KD, organ dysfunctions, treatment, and outcomes. Laboratory data were hemoglobin, total white blood cell (WBC) count, absolute lymphocyte count (ALC), platelet count, erythrocyte sedimentation rate (ESR), CRP, aspartate transaminase (AST), alanine transaminase (ALT), albumin, troponin (Tn), N-terminal pro-brain natriuretic peptide (NT-proBNP), ferritin, triglycerides (TG), fibrinogen, international normalized ratio (INR), and pyuria (≥10 WBC/high-power field). Whether patients met the 2016 macrophage activation syndrome (MAS) criteria [23] was also assessed.

Microbiological results were recorded, including bacterial or viral cultures, serology, and polymerase chain reaction (PCR) analysis. For patients with MIS-C, the date of COVID-19 diagnosis before hospitalization and the results of SARS-CoV-2 PCR tests, SARS-CoV-2-specific antigen, and SARS-CoV-2-specific antibodies during hospitalization were investigated separately. When calculating the incidence of KDSS and MIS-C, KD patients who received treatment with IVIG (2 g/kg/dose) and aspirin (30–50 mg/kg/day) during hospitalization were included in the number of KD patients.

### 2.4. Statistical Analysis

Data for categorical variables are presented as frequencies and percentages, while continuous variables are displayed as medians and ranges. For statistical analysis, Fisher’s exact test was used to compare categorical variables and the Mann–Whitney *U* test was used to compare continuous variables. All tests were two-tailed, and a *p*-value < 0.05 was considered statistically significant. Statistical analyses were performed using IBM SPSS Statistics software Version 24.0 (IBM Corporation, Armonk, NY, USA).

## 3. Results

### 3.1. Incidence of KDSS and MIS-C

Thirty-one patients were enrolled in this study; twenty-two met the case definition of MIS-C and nine met the definition of KDSS. The study period was divided into the “pre-COVID-19 pandemic” (January 2013 to December 2019) and the “COVID-19 pandemic” (January 2020 to December 2022). The total number of KD patients was 1095: 879 (125 per year) during the pre-COVID-19 pandemic and 216 (72 per year) during the COVID-19 pandemic. 

The annual (first- and second-half) incidences of KDSS and MIS-C compared with KD are illustrated in Figure 1. The incidence of KDSS in KD was 0.8% (9/1095): 0.8% (7/879) during the pre-COVID-19 pandemic and 0.9% (2/216) during the COVID-19 pandemic. The incidence of MIS-C versus KD was 10.2% (22/216) during the COVID-19 pandemic.

### 3.2. Demographic and Clinical Characteristics

Of the 31 patients enrolled, the median age was 9.3 years (0.9–16.0) and the male-to-female ratio was 0.94 (15/16). The median height was 135.0 cm (70.0–182.0), the median weight was 27.2 kg (8.8–72.6), the median length of hospital stay was 10.0 days (4.0–57.0), and the median duration of fever at admission was 5.0 days (2.0–15.0). Most enrolled patients met the KD diagnostic criteria (90.3%), and conjunctivitis (83.9%) and rash (80.6%) were the commonly observed KD-like features. 

Comparing the demographic and clinical characteristics of the two groups, the KDSS group had a longer hospital stay than the MIS-C group (*p* = 0.007), and there were no significant differences in other variables (Table 1).

### 3.3. Laboratory Characteristics

Among the 31 enrolled patients, the following laboratory abnormalities were observed: anemia (35.5%), leukocytosis or lymphopenia (71.0%), thrombocytopenia (58.1%), CRP elevation (100.0%), elevated liver transaminases (83.9%), hypoalbuminemia (71.0%), coagulopathy (74.2%), and adherence to the 2016 MAS criteria (48.4%). Both groups showed severe systemic inflammation, but the overall severity of inflammation tended to be more severe in the KDSS group than in the MIS-C group. 

Table 2 shows the laboratory characteristics of the two groups. In comparison with the MIS-C group, the KDSS group exhibited significantly lower median values of hemoglobin (*p* = 0.007) and albumin (*p* = 0.011), along with a higher median value of CRP (*p* = 0.022). Moreover, pyuria was more prevalent in the KDSS group than in the MIS-C group (*p* = 0.006). The KDSS group also exhibited higher median values of WBC count, ESR, AST, ALT, NT-proBNP, and ferritin, along with lower median values of ALC and platelet count compared with the MIS-C group (*p* > 0.05). Coagulopathy was more prevalent in the KDSS group than in the MIS-C group, but abnormal TG or fibrinogen and meeting the 2016 MAS criteria were more common in the MIS-C group than in the KDSS group (*p* > 0.05).

### 3.4. Organ Dysfunction According to the CSTE/CDC Definition

All 31 patients in the MIS-C and KDSS groups had two or more organ-dysfunction events, with a median of 4.0 organs (2.0–5.0) involved. The frequencies of cardiac, shock, mucocutaneous, gastrointestinal, or hematologic dysfunction were 9 (61.3%), 15 (48.4%), 30 (96.8%), 28 (90.3%), and 23 (74.2%), respectively. Additionally, 11 patients (35.5%) experienced other organ dysfunctions, such as neurologic, renal, or musculoskeletal manifestations.

Table 3 presents the four components of the CSTE/CDC definition and demonstrates their applicability to the MIS-C and KDSS groups. In both groups, all patients were febrile children or adolescents aged <21 years, had systemic inflammation (CRP ≥ 3.0 mg/dL), and showed two or more organ-dysfunction events. In a comparison of organ dysfunction between the two groups, the median number of organs involved was higher in the KDSS group than in the MIS-C group (*p* = 0.040), as was the frequency of shock (*p* < 0.001). All patients in the MIS-C group, but none in the KDSS group, had laboratory evidence of SARS-CoV-2 infection (*p* < 0.001). Of the nine patients in the KDSS group, seven were hospitalized during the pre-COVID-19 pandemic and two were hospitalized during the pandemic but had no history of recent SARS-CoV-2 infection and tested negative for SARS-CoV-2 PCR, SARS-CoV-2-specific antigen, and SARS-CoV-2-specific antibodies during hospitalization.

### 3.5. Treatment and Outcomes

For the initial treatment, all patients in both groups received either IVIG or systemic steroids or both. Specifically, 15 (48.4%) received both IVIG and systemic steroids, 12 (38.7%) received IVIG alone, and 4 (12.9%) received systemic steroids alone. Of the 31 patients, 11 (35.4%) received second-line treatment due to persistent fever after the initial treatment with IVIG ± systemic steroids. For the second-line treatment, 11/11 (100.0%) received systemic steroids, 7/11 (63.6%) received a second IVIG infusion, and 2/11 (18.2%) received a biologic or other drug (i.e., anakinra, tocilizumab, infliximab, cyclosporin, or methotrexate). 

Empirical antibiotics were administered to 28 patients (87.1%) and vasoactive drugs were given to 13 (41.9%). ICU admission was required in 14 (45.2%) and CAAs were observed in 9 (29.0%). Notably, all 31 patients in both groups recovered successfully. Comparing the treatment and outcomes between the two groups, the frequencies of vasoactive infusion (*p* = 0.001) and ICU admission (*p* < 0.001) were higher in the KDSS group than in the MIS-C group. Comparisons of other variables of treatment and outcomes revealed no statistical differences (Table 4).

## 4. Discussion

In this study, the incidence of KDSS in KD was 0.8% and the incidence of MIS-C versus KD was 10.2%. All 31 patients included in the MIS-C and KDSS groups presented with KD-like features, systemic inflammation, and organ dysfunction. The severity of systemic inflammation and the extent of organ dysfunction were more pronounced in the KDSS group than in the MIS-C group. All in the MIS-C group had laboratory evidence of SARS-CoV-2 infection, but none in the KDSS group.

Common presentations of MIS-C include prolonged fever, KD-like features, gastrointestinal symptoms, shock, or coagulopathy [1,2,3]. Many of these features are also encountered in cases of acute abdomen, toxic shock syndrome (TSS), septic shock, or multi-organ dysfunction syndrome (MODS), which sometimes causes diagnostic confusion in clinical practice [12,14]. In the first case report of MIS-C in Korea [24], the patient showed abdominal pain, shock, and heart failure without KD-like features and was initially considered to have septic shock of gastrointestinal origin. Conjunctivitis and oropharyngeal inflammation appeared on the sixth day of hospitalization. Because KD-like features may not be present in the acute phase of MIS-C, and one-quarter of patients with MIS-C do not develop KD-like features throughout their disease [25,26], MIS-C should be suspected in patients presenting with unknown etiology of shock, heart failure, or gastrointestinal symptoms, even in the absence of KD-like features [10,27].

Although KDSS is a subtype of KD, the two are significantly different [11,12]. In KDSS, systemic inflammatory markers, such as anemia, thrombocytopenia, CRP elevation, hypoalbuminemia, or hyperferritinemia, are more frequent and more severe compared with KD [15]. While some patients with KD may develop rare complications of gastrointestinal, neurologic, or musculoskeletal manifestations, KDSS is characterized by the presence of multi-organ dysfunction, including cardiac, gastrointestinal, hematologic, neurologic, renal, and musculoskeletal manifestations [16]. Epidemiologically, KDSS also shows distinct characteristics [17]. Compared with KD patients, those with KDSS have a higher median age, and KDSS is more common in non-Asian regions and includes a larger Hispanic population [13,14]. Based on these differences, it had been suggested that KDSS should be classified as a distinct disease entity from KD [15]. As with MIS-C, KDSS often presents with gastrointestinal symptoms with severe inflammation, and it can be mistaken as acute abdomen, TSS, septic shock, or MODS [11,12]. KDSS and MIS-C may present a variety of clinical manifestations, and delayed recognition of the disease can result in a fatal outcome [14,18].

Table 5 compares the important characteristics of MIS-C, KDSS, and KD, utilizing data from a United States study on MIS-C (*n* = 186) [26], a literature review on KDSS (*n* = 103) [13], and a Korean nationwide survey on KD (*n* = 14,916) [28]. Unlike KD, both MIS-C and KDSS predominantly affect patients older than five years [18]. Whereas KD is frequent in East Asia (Japan, Korea, and Taiwan), both MIS-C and KDSS are relatively frequent in Europe and North America (Hispanic and black populations) and rare in East Asia (Asian populations) [17]. All three diseases present with systemic inflammation and organ dysfunction. However, the overall severity, such as duration of fever, cardiac dysfunction, IVIG resistance, CAAs, or mortality, follows the order of KDSS > MIS-C > KD. Non-cardiac organ dysfunction is also frequent in MIS-C and KDSS but rare in KD. Based on the epidemiological, clinical, and immunological differences between MIS-C and KD, Sharma et al. [8] proposed that the two be considered different diseases. Irrespective of the relationships among MIS-C, KDSS, and KD, there is no doubt that MIS-C shares more overlapping characteristics with KDSS than it does with KD.

In this study, the incidence of KDSS (0.8%, 9/1095) was similar to that reported in Japan (1.1%, 6/552) [14] but much lower than that in the United States (7.0%, 13/187) [15]. The incidence of MIS-C in the present study (10.2%, 22/216) was less than 1/16 of that in the United States (166.3%, 31.6/19.0) [29]. The United States data were calculated from the incidence of MIS-C during the COVID-19 pandemic (0.032%, 31.6 cases per 100,000 children with COVID-19) and the incidence of KD before the COVID-19 pandemic (0.019%, 19.0 cases per 100,000 children aged <5 years). Compared with the literature review on KDSS [13], the KDSS patients in this study were older (9.3 years vs. 5.2 years), exhibited more severe manifestations (e.g., thrombocytopenia, elevated liver transaminases, or coagulopathy), and more frequently received vasoactive drugs (88.9% vs. 66.9%) and systemic steroids (77.8% vs. 35.9%). Compared with the patients in the United States study [26], the MIS-C patients in this study exhibited lower frequencies of shock (27.3% vs. 48.9%) and vasoactive infusion (22.7% vs. 48.9%). Briefly, a lower incidence of MIS-C and KDSS, lower severity of MIS-C, and higher severity of KDSS were found in this study compared with data from the two previous studies [13,26].

Figure 2 illustrates the relationships between KD, KDSS, and MIS-C. KD is characterized by KD-like features included in the KD diagnostic criteria and systemic inflammation [21]. As an infectious trigger, a broad spectrum of microbes have been identified in patients with KD and are considered to play an important role in the pathogenesis of KD (Figure 2A) [30,31,32]. KDSS is diagnosed when shock as organ dysfunction is observed in patients with KD-like features and systemic inflammation. A diagnosis of KDSS does not require evidence of concomitant infections, but several microbes such as Epstein Barr virus, influenza virus, or group A streptococcus have been found in KDSS patients (Figure 2B) [10,33]. MIS-C is diagnosed when SARS-CoV-2 is identified in patients with KD-like features, systemic inflammation, and organ dysfunction such as shock (Figure 2C) [30]. 

Among the KDSS group of nine patients, two were hospitalized during the COVID-19 pandemic. Although both patients presented with severe systemic inflammation and multi-organ dysfunction, there was no evidence of recent SARS-CoV-2 infection in their medical history and microbiological examinations. They were diagnosed with KDSS. If evidence of SARS-CoV-2 infection had been demonstrated in these two KDSS patients, they should have been diagnosed with MIS-C. Similarly, if evidence of SARS-CoV-2 infection could not be demonstrated in patients with suspected MIS-C due to KD-like features and shock, they should have been diagnosed with KDSS. In other words, the most important difference between MIS-C and KDSS is whether SARS-CoV-2 has been identified as an infectious trigger [10].

MAS is a potentially fatal complication that can be overlooked in KD-like hyperinflammatory diseases, such as KD, KDSS, other severe forms of KD, or MIS-C [8,9]. The reported incidence of MAS was 1.9% (12/638) for KD, 3.2% (8/247) for incomplete KD, 4.8% (3/63) for KDSS, and 6.3% (4/63) for refractory KD [22]. In patients with MIS-C, the incidence of MAS has been reported to be much higher at 22.7% (5/22) [6], 42.1% (8/19) [9], and 50.0% (5/10) [3]. Rodriguez-Smith et al. [9] reported that patients with MIS-C who met the 2016 MAS criteria had significantly higher levels of interferon-related chemokines than did patients with MIS-C who did not meet the 2016 MAS criteria. In the present study, 50.0% (11/22) of the MIS-C group and 44.4% (4/9) of the KDSS group met the 2016 MAS criteria. Given the life-threatening clinical course of MAS, early diagnosis in KD-like hyperinflammatory diseases, including MIS-C and KDSS, is essential [33]. When patients with MIS-C or KDSS show unexpected clinical deterioration despite appropriate treatment, screening tests, including those that measure ferritin level, should be performed to rule out MAS [34,35,36].

This relatively small and retrospective study has some limitations, and a prospectively designed large-scale study is needed to validate our findings. The number of patients with KDSS could have been underestimated because KD patients who received vasoactive drugs or whose secondary diagnosis was hypotension were selected and included in the KDSS group. This could explain the relatively low incidence of KDSS and the relatively high severity of KDSS in this study.

## 5. Conclusions

All patients diagnosed with MIS-C or KDSS in this study exhibited KD-like features, systemic inflammation, and organ dysfunction. The overall severity of systemic inflammation and organ dysfunction was higher in KDSS patients than in those with MIS-C. MIS-C is diagnosed when SARS-CoV-2 infection is demonstrated in a patient with KD-like features, systemic inflammation, and organ dysfunction such as shock. Otherwise, KDSS is diagnosed. In other words, the fundamental difference between MIS-C and KDSS is whether SARS-CoV-2 has been identified as an infectious trigger. MIS-C shares more overlapping characteristics with KDSS than with KD. Understanding the similarities and differences between MIS-C and KDSS will provide useful clues for studying the pathogenesis and therapeutic strategies of KD-like hyperinflammatory diseases, including MIS-C and KDSS.

## Figures and Tables

**Figure 1 children-10-01527-f001:**
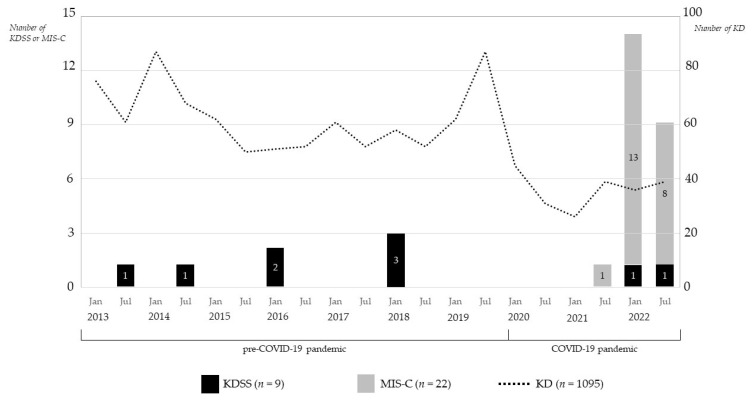
The annual (first- and second-half) incidences of KDSS and MIS-C compared with KD. A total of 9 patients with KDSS were hospitalized during the 10-year study period (i.e., pre-COVID-19 pandemic + COVID-19 pandemic) and 22 patients with MIS-C were hospitalized during the COVID-19 pandemic. Dividing by the number of patients with KD, the incidence of KDSS was 0.8% (9/1095) and the incidence of MIS-C was 10.2% (22/216). COVID-19, coronavirus disease 2019; KDSS, Kawasaki disease shock syndrome; MIS-C, multisystem inflammatory syndrome in children; KD, Kawasaki disease.

**Figure 2 children-10-01527-f002:**
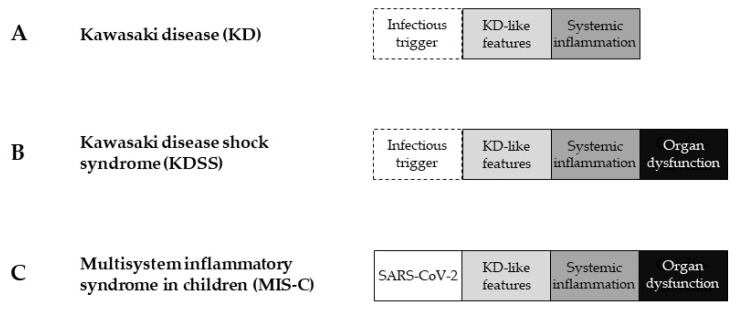
Relationships between KD, KDSS, and MIS-C. (**A**) KD is characterized by KD-like features and systemic inflammation. Infectious triggers are expected to play an important role in the pathogenesis of KD. (**B**) KDSS is diagnosed when shock presenting as organ dysfunction is observed in patients with KD-like features and systemic inflammation. (**C**) MIS-C is diagnosed when SARS-CoV-2 is identified in patients with KD-like features, systemic inflammation, and organ dysfunction such as shock. SARS-CoV-2, severe acute respiratory syndrome coronavirus 2; KD, Kawasaki disease; KDSS, Kawasaki disease shock syndrome; MIS-C, multisystem inflammatory syndrome in children.

**Table 1 children-10-01527-t001:** Comparison of demographic and clinical characteristics between the two groups.

	MIS-C (*n* = 22)	KDSS (*n* = 9)	*p*-Value
Age, years	9.3 (0.9–16.0)	9.3 (1.0–15.5)	0.965
Male sex	11 (50.0)	4 (44.4)	1.000
Height, cm	135.3 (74.5–182.0)	132.0 (70.0–159.0)	0.632
Weight, kg	31.3 (9.9–72.6)	32.0 (8.8–47.0)	0.679
Length of hospital stay, days	10.0 (4.0–18.0)	11.0 (8.0–57.0)	0.007
Duration of fever at admission, days	5.0 (3.0–15.0)	5.0 (2.0–15.0)	0.606
Rash	17 (77.3)	8 (88.9)	0.642
Conjunctivitis	18 (81.8)	8 (88.9)	1.000
Oropharyngeal inflammation	10 (45.5)	6 (66.7)	0.433
Extremity changes	13 (59.1)	4 (44.4)	0.693
Cervical lymphadenopathy	15 (68.2)	4 (44.4)	0.253
Complete KD	11 (50.0)	5 (55.6)	1.000
Met KD criteria	19 (86.4)	9 (100.0)	0.537

Data are presented as frequency (%) or median (range). MIS-C, multisystem inflammatory syndrome in children; KDSS, Kawasaki disease shock syndrome; KD, Kawasaki disease.

**Table 2 children-10-01527-t002:** Comparison of laboratory characteristics between the two groups.

	MIS-C (*n* = 22)	KDSS (*n* = 9)	*p*-Value
Hemoglobin, g/dL	12.1 (9.6–14.9)	9.5 (8.6–13.1)	0.007
Total WBC count, 10^3^/μL	7.2 (2.3–20.4)	27.5 (3.3–37.4)	0.139
ALC, μL	937 (198–6890)	912 (163–4368)	0.728
Platelet count, 10^3^/μL	130 (62–431)	114 (59–414)	0.931
ESR, mm/h	26 (2–120)	43 (13–92)	0.372
CRP, mg/dL (normal < 0.5)	10.1 (3.2–25.5)	20.5 (3.5–26.7)	0.022
AST, U/L	55 (33–579)	75 (24–299)	0.223
ALT, U/L	54 (13–232)	65 (15–236)	0.557
Albumin, g/dL	2.9 (2.4–4.1)	2.5 (2.3–3.3)	0.011
NT-proBNP, pg/dL	1464 (299–23,858)	12,136 (813–55,322)	0.065
Ferritin, ng/mL	566 (158–10,364)	756 (218–5130)	0.907
Abnormal TG or fibrinogen ^a^	17 (77.3)	4 (44.4)	0.105
Coagulopathy: INR > 1.1	15 (68.2)	8 (88.9)	0.379
Pyuria: ≥10 WBC/hpf	7 (31.8)	8 (88.9)	0.006
Met 2016 MAS criteria ^b^	11 (50.0)	4 (44.4)	1.000

Data are presented as frequency (%) or median (range). MIS-C, multisystem inflammatory syndrome in children; KDSS, Kawasaki disease shock syndrome; WBC, white blood cell; ALC, absolute lymphocyte count; ESR, erythrocyte sedimentation rate; CRP, C-reactive protein; AST, aspartate transaminase; ALT, alanine transaminase; NT-proBNP, N-terminal pro-brain natriuretic peptide; TG, triglyceride; INR, international normalized ratio; hpf, high-power field; MAS, macrophage activation syndrome. ^a^ TG > 156 mg/dL or fibrinogen ≤ 360 mg/dL. ^b^ MAS is diagnosed when ferritin > 684 ng/mL plus 2 of 4 criteria (platelets ≤ 181,000/μL, AST > 48 U/L, TG > 156 mg/dL, or fibrinogen ≤ 360 mg/dL) [23].

**Table 3 children-10-01527-t003:** CSTE/CDC case definition [7] and its application to the two groups.

	MIS-C (*n* = 22)	KDSS (*n* = 9)	*p-*Value
1. A patient aged <21 years with fever (≥38.0 °C)	22 (100.0)	9 (100.0)	1.000
2. Systemic inflammation: CRP ≥ 3.0 mg/dL	22 (100.0)	9 (100.0)	1.000
3. Organ dysfunction ≥ 2/5 categories			
Cardiac: EF < 55%, CAAs, or elevated Tn	12 (54.5)	7 (77.8)	0.418
Shock: clinician documentation in medical records	6 (27.3)	9 (100.0)	<0.001
Mucocutaneous: rash, conjunctivitis, oropharyngeal	21 (95.5)	9 (100.0)	1.000
inflammation, or extremity changes			
Gastrointestinal: vomiting/diarrhea or abdominal pain	20 (90.9)	8 (88.9)	1.000
Hematologic: platelet < 150,000/μL or ALC < 1000/μL	17 (77.3)	6 (66.7)	0.660
Other organs: neurologic, renal, or musculoskeletal ^a^	6 (27.3)	5 (55.6)	0.217
Number of organs involved	3.0 (2.0–5.0)	4.0 (3.0–5.0)	0.040
4. Laboratory evidence of SARS-CoV-2 infection	22 (100.0)	0 (0.0)	<0.001

Data are presented as frequency (%) or median (range). CSTE/CDC, Council of State and Territorial Epidemiologists/Centers for Disease Control and Disease Prevention; MIS-C, multisystem inflammatory syndrome in children; KDSS, Kawasaki disease shock syndrome; CRP, C-reactive protein; EF, ejection fraction; CAA, coronary artery abnormalities; Tn, troponin; ALC, absolute lymphocyte count; SARS-CoV-2, severe acute respiratory syndrome coronavirus 2. ^a^ Neurologic, renal, or musculoskeletal involvements are not included in the CSTE/CDC case definition.

**Table 4 children-10-01527-t004:** Comparison of treatment and outcomes between the two groups.

	MIS-C (*n* = 22)	KDSS (*n* = 9)	*p*-Value
Initial IVIG	19 (86.4)	8 (88.9)	1.000
IVIG resistance (initial treatment failure)	7 (31.8)	4 (44.4)	0.683
Second IVIG	4 (18.2)	3 (33.3)	0.384
Systemic steroids	20 (90.9)	7 (77.8)	0.560
Biologics or other drugs ^a^	1 (4.5)	1 (11.1)	0.503
Empirical antibiotics	19 (86.4)	9 (100.0)	0.537
Vasoactive drugs	5 (22.7)	8 (88.9)	0.001
ICU admission	5 (22.7)	9 (100.0)	<0.001
Coronary artery complications	5 (22.7)	4 (44.4)	0.385
Mortality	0 (0.0)	0 (0.0)	1.000

Data are presented as frequency (%). MIS-C, multisystem inflammatory syndrome in children; KDSS, Kawasaki disease shock syndrome; IVIG, intravenous immunoglobulin; ICU, intensive care unit. ^a^ Biologics or other drugs include anakinra, tocilizumab, infliximab, cyclosporin, and methotrexate.

**Table 5 children-10-01527-t005:** Important characteristics of MIS-C [26], KDSS [13], and KD [28] ^a^.

	MIS-C, US (*n* = 186)	KDSS, Literature Review (*n* = 103)	KD, Korea (*n* = 14,916)
Demographic and clinical			
Age, years	8.3	5.2	2.4
Male sex, %	61.8	55.3	58.4
Race and ethnic group	Hispanic or black	Hispanic	Asian
Duration of fever, days	6.0	7.2	5.7
Complete KD, %	39.8	69.9	67.2
Laboratory abnormalities, %			
Anemia	47.6	58.8	–
Leukocytosis or lymphopenia	79.9	64.4	–
CRP elevation	90.7	91.9	–
Elevated liver transaminases	63.7	61.4	–
Hypoalbuminemia	80.3	93.4	–
Coagulopathy: INR > 1.1 or prolonged PT	77.0	54.8	–
Organ dysfunction, %			
Cardiac: EF < 55%, CAAs, or elevated Tn	50.3	65.0	10.8
Shock	48.9	100.0	–
Mucocutaneous	73.6	100.0	100.0
Gastrointestinal	91.9	74.6	26.4 ^b^
Hematologic: thrombocytopenia	54.6	79.5	–
Other organs: neurologic, renal, or musculoskeletal	37.6	53.9	5.5 ^b^
Treatment and outcomes, %			
Initial IVIG	77.4	95.1	95.4
IVIG resistance (initial treatment failure)	41.4	47.6	11.8
Second IVIG	21.0	34.0	8.4
Systemic steroids	48.9	35.9	3.0
Biologics or other drugs ^c^	20.4	16.5	0.005
Vasoactive drugs	48.9	66.9	–
Coronary artery complications	8.8 ^d^	65.0	10.8
Mortality	2.1 (4 cases)	6.8 (7 cases)	0.0001 (2 cases)

MIS-C, multisystem inflammatory syndrome in children; KDSS, Kawasaki disease shock syndrome; KD, Kawasaki disease; – not available; CRP, C-reactive protein; INR, international normalized ratio; PT, prothrombin time; EF, ejection fraction; CAAs, coronary artery abnormalities; Tn, troponin; IVIG, intravenous immunoglobulin. ^a^ Modified from our previous study with permission [10]. ^b^ Data from [16]. ^c^ Biologics or other drugs include anakinra, tocilizumab, infliximab, cyclosporin, and methotrexate. ^d^ The frequency was much lower than expected because only severe lesions (i.e., z ≥ 2.5) were considered coronary artery complications in their study.

## Data Availability

Data from this study can be obtained upon request to the authors.

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
