# Peer review of "Similarities and Differences between Multisystem Inflammatory Syndrome in Children (MIS-C) and Kawasaki Disease Shock Syndrome"

_children, 2023, doi:10.3390/children10091527_

Round 1

Reviewer 1 Report

The article described the problem in wide way. The methodology is quite good. I see only the few mistakes, which shoul be improved. 

Line 142 mistake in description

Line 143 mistake in description

Line 152. You write that the time of fever was 2-15 days. 2 days of fever are not incuded in Kawasaki D criteria. 

Line 172 - unify the shortages description ( They should be the same like in text)

Line 277- In literature review, The European articles should be also described. Not only American.

Ludwikowska, K.M., Okarska-NapieraÅ‚a, M., Dudek, N. et al. Distinct characteristics of multisystem inflammatory syndrome in children in Poland. Sci Rep 11, 23562 (2021). https://doi.org/10.1038/s41598-021-02669-2

Schlapbach LJ, Andre MC, Grazioli S, et al. Best Practice Recommendations for the Diagnosis and Management of Children With Pediatric Inflammatory Multisystem Syndrome Temporally Associated With SARS-CoV-2 (PIMS-TS; Multisystem Inflammatory Syndrome in Children, MIS-C) in Switzerland. Front Pediatr. 2021;9:667507. Published 2021 May 26. doi:10.3389/fped.2021.667507

Non published materials are sent in Chineese, I can not read it. In my opinion they should be in English. 

The English language is correct

Author Response

Response to Reviewer 1 Comments

Point 1: Line 142 mistake in description

Response 1: We apologize for any typos.

‘pa-tients’ → ‘patients’

Point 2: Line 143 mistake in description

Response 2: We apologize for any typos.

‘pan-demic’ → ‘pandemic’

Point 3: Line 152. You write that the time of fever was 2-15 days.

2 days of fever are not incuded in Kawasaki D criteria.

Response 3: As stated in Line 152, ‘duration of fever’ means ‘duration of fever at the time of admission’, not  ‘duration of fever at the time of diagnosis.’

To avoid confusion, ‘duration of fever (or fever duration)’ in the main text and Table 1 was changed to ‘duration of fever at admission.’

Point 4: Line 172 - unify the shortages description (They should be the same like in text)

Response 4: The lack of statistical significance has been denoted as ‘p >0.05’.

Here is a corrected version of the sentence:

“The KDSS group also exhibited higher median values of WBC count, ESR, AST, ALT, NT-proBNP, and ferritin, along with lower median values of ALC and platelet count compared with the MIS-C group (p > 0.05). Coagulopathy was more prevalent in the KDSS group than in the MIS-C group, but abnormal TG or fibrinogen and meeting the 2016 MAS criteria were more common in the MIS-C group than in the KDSS group (p > 0.05).

Point 5: In literature review, The European articles should be also described. Not only American.

Response 5: We completely aggree with you a literature review on MIS-C should covered both European and American articles. However, Table 5 is not intended to be a literature review on MIS-C, KDSS, and KD.

* We selected three simple and general articles on MIS-C, KDSS, and KD and roughly compared their characteristics in Table 5. The MIS-C study (Ref. 27) was conducted in the US, and the KD study (Ref. 29) conducted in Korea. The KDSS study through literature review (Ref. 13) was inevitably chosen because KDSS itself is a very rare disease.

* The revision includes two articles recommended by Reviewer 1 as references.

Point 6: Non published materials are sent in Chineese, I can not read it. In my opinion they should be in English

Response 6: Non-published materials includes the documents bearing the authors’ name and signature (not in Enghish). So there is no need to read or understand that documents. We apologize for the inconvenience.

Reviewer 2 Report

Comments: Electrocardiograph results.

Comment: Values of Z-scores.

Comment: Myocarditis, valvular abnormalities.

Comment: Table 3 describe Coronary artery abnormalities.

Comment: Table 4 describe Coronary artery complications.

Comments: Extended follow-up on both groups with cardiac dysfunction, especially those with CAA.

Regarding reference 3, the volume is 395 and should go together with Riphagen S. Lancet 2020; 395:1607.

References that should be reviewed and commented on:

Kawasaki T. Pediatrics 1974; 54:271.

Dufort EM. N Engl J Med 2020; 383:347.

Godfred-Cato S. Pediatr Infect Dis J 2022; 41:315.

None

Author Response

Response to Reviewer 2 Comments

Point 1: Commnet: Electrocardiograph results.

Response 1: ECG is an essential test for MIS-C, KDSS, and KD. As stated in Line 76 (Method section), the study data were collected through the Clinical Data Warehouse (CDW). Unfortunately, CDW does not provide images of the ECG.

Point 2: Comment: Values of Z-scores

Response 2: Among the 31 enrolled patients, 9 patients had CAAs. Of the 9 patients with CAAs, 5 had a z-score ≥ 2.5 (coronary artery aneurysms) and 4 had a z-score ≥ 2.0 (coronary artery dilatation). CAAs were defined in Line 104-105 (Methods section).

Point 3: Comment: Myocarditis, valvular abnormalities.

Response 3: On echocardiography, 19.3% (6/31) showed myocariditis, 19.3% (6/31) showed valvular abnormalities, and 16.1% (5/31) showed both myocariditis and valvular abnormalities.

Point 4: Comment: Table 3 describe Coronary artery abnormalities.

Response 4: Table 3 compares organ dysfunction between the two diseases applying the CSTE/CDC case definitions. This definition divides organ dysfunction into five categories: cardiac, shock, mucocutaneous, gastrointestinal, and hematologic. Therefore, CAAs cannot be described separately in Table 3 and are included in the category of cardiac dysfunction.

Point 5: Comment: Table 4 describe Coronary artery complications.

Response 5: Among the 31 enrolled patients, 9 patients had CAAs. Of the 9 patients with CAAs, 5 had coronary artery aneurysms (z-score ≥ 2.5) and 4 had coronary artery dilatation (z-score ≥ 2.0).

Point 6: Extended follow-up on both groups with cardiac dysfunction, especially those with CAA.

Response 6: Seven out of nine patients with CAA fully recovered within one year. No improvement in CAA was observed in one patient of the KDSS groups and one patient of the MIS-C group.

Response 2-6: The aim of this study was to compare the “general characteristics” between MIS-C and KDSS, including the presence of KD-like clinical features, the degree of systemic inflammation, and overall organ dysfunction. Cardiac involvement (e.g., Z-score, myocarditis, and valvualr abmormalities), a key feature of MIS-C and KDSS will be the discussed in detail in the next article. Some of the co-authors are pediatric cardiologists (see Ref. 19).

Point 7: Regarding reference 3, the volume is 395 and should go together with Riphagen S. Lancet 2020; 395:1607.

Response 7: We apologize for any typos; ‘295’ → ‘395’

Riphagen et al.’s article is included as a new reference in the revision.

Point 8: References that should be reviewed and commented on:

Response 8: The revision includes three articles recommended as references by Reviewer 2.

Reviewer 3 Report

The study compared COVID-19-associated multisystem inflammatory syndrome in children (MIS-C) and Kawasaki disease shock syndrome (KDSS), finding that both shared various characteristics. This is a manuscript with a clearly written aim and conclusion.

Minor comments:

Line 46 48: Please explain if there is a difference between MIS-C and PIMS-TS.

Line 142: Should be “patients” instead of “pa-tients”.

Author Response

Response to Reviewer 3 Comments

Point 1: Line 46-48: Please explain if there is a difference between MIS-C and PIMS-TS.

Response 1: MIS-C is characterized by hyperinflammatory and multiple organ involvement and usually occurs 2–6 weeks after the acute phase of SARS-CoV-2 infection. This syndrome is called PIMS-TS in Europe and MIS-C in North America.

To avoid confusion, the World Health Organization recommend naming this syndrome MIS-C.

In other words, MIS-C and PIMS-TS are considered synonymous.

Point 2: Line 142: Should be “patients” instead of “pa-tients”.

Response 2: We apologize for any typos.

‘pa-tients’ → ‘patients’
